# Protective Effect of *Bifidobacterium animalis* subs. *lactis* MG741 as Probiotics against UVB-Exposed Fibroblasts and Hairless Mice

**DOI:** 10.3390/microorganisms10122343

**Published:** 2022-11-26

**Authors:** Ji Yeon Lee, Jeong-Yong Park, YongGyeong Kim, Chang-Ho Kang

**Affiliations:** MEDIOGEN, Co., Ltd., Biovalley 1-ro, Jecheon-si 27159, Republic of Korea

**Keywords:** *Bifidobacterium animalis* ssp. *lactis* MG741, matrix metalloproteinases, anti-photoaging, ultraviolet B

## Abstract

Skin photoaging, which causes wrinkles, increased epidermal thickness, and rough skin texture, is induced by ultraviolet B (UVB) exposure. These symptoms by skin photoaging have been reported to be involved in the reduction of collagen by the expression of matrix metalloproteinases (MMPs) and activator protein-1 (AP-1). This study investigated the protective effects of *Bifidobacterium animalis subsp. lactis* MG741 (*Bi. lactis* MG741) in Hs-68 fibroblasts and hairless mice (HR-1) following UVB exposure. We demonstrated that the *Bi. lactis* MG741 reduces wrinkles and skin thickness by downregulating MMP-1 and MMP-3, phosphorylation of extracellular signal-regulated kinase (ERK), and c-FOS in fibroblasts and HR-1. Additionally, in UVB-irradiated dorsal skin of HR-1, *Bi. lactis* MG741 inhibits the expression of nuclear factor kappa-light-chain-enhancer of activated B cells (NF-κB), an inflammation-related factor. Thus, *Bi. lactis* MG741 has the potential to prevent wrinkles and skin inflammation by modulating skin photoaging markers.

## 1. Introduction

The skin is the largest organ in the body accounting for 16% of the body weight [1]. The skin acts as a barrier to protect against various external stimuli, such as physical stress, smoking, air pollution, pathogens, and ultraviolet (UV) radiation [2]. Since the skin is the most exposed organ of the body externally, its aging can be caused by stimulation by environmental factors, such as UV, which is the most harmful external factor [3]. Although most UV radiation is absorbed by the epidermis of the skin, ultraviolet B (UVB) affects dermal fibroblasts by activating factors, including activator protein-1 (AP-1) and nuclear factor kappa-light-chain-enhancer of activated B cells (NF-κB), which are related to aging in the epidermis [4]. UVB irradiation in the skin causes skin aging, such as wrinkles, inflammation, and even leading to skin cancer [5,6]. Therefore, it is crucial to develop agents that are effective for photoaging having are proven safe [7].

It has been demonstrated that the formation of wrinkles in the skin is caused by the breakdown of collagen as an activation mechanism of matrix metalloproteinases (MMPs) due to external factors of aging, such as UVB exposure [8]. The regulation of MMPs production is mainly affected by AP-1 activation stimulated by the mitogen-activated protein kinases (MAPKs) [9]. Moreover, UVB exposure increases inflammatory factors, including NF-κB, interleukin-6 (IL-6), and tumor necrosis factor-α (TNF-α), which upregulate MMPs activation in the skin [5,9].

According to the World Health Organization (WHO) and Food and Agriculture Organization of the United Nations (FAO), probiotics, including *Lactobacillus* and *Bifidobacterium*, are living microorganisms that have health benefit on host, when administered in adequate amounts [10,11]. Probiotics, as dietary supplements, are generally known to have a protective effect against gastrointestinal disorders [12]. Probiotics has been actively studied for skin diseases such as atopic dermatitis, psoriasis, acne, and photoaging [13]. It has been previously reported that *Lactobacillus plantarum*, *Bi. longum*, and *Bi. breve* have shown efficacy in preventing photo-aging [13]. Metabolites contained in the cell-free supernatant (CFS) of probiotics that affect skin health include diacetyl, lipoteichoic acid, lactic acid, and acetic acid [14]. Probiotics have recently been used as potential agents with various health benefits and are attracting attention as an effective alternative to skin aging [15].

Oxidative stress caused by reactive oxygen species (ROS) in the skin is caused by UVB, which can exacerbate the skin-aging [16]. To prevent oxidative stress in the skin, various antioxidants, such as vitamin polyphenols and collagen peptides, have been applied [17]. We have previously demonstrated that *Bi. lactis* MG741 has antioxidant potential in a tert-butyl hydroperoxide (t-BHP)-induced animal model, and was evaluated for safety as probiotics [18]. However, the underlying mechanisms for intrinsic and extrinsic aging of the skin associated with the protective effect of MG741 against UVB have not been elucidated. Therefore, this study demonstrates the protective effects of *Bi. lactis* MG741 against UVB-induced photoaging in Hs68 fibroblasts and hairless (HR-1) mice, and investigated the potential mechanisms. 

## 2. Materials and Methods

### 2.1. Apparatus and Reagents

For UVB irradiation, a UV crosslinker was purchased from BoTeck (Gunpo, Republic of Korea) for cell assay. A UVB lamp was purchased from Philips (Amsterdam, Netherlands) for the animal study. A microplate spectrophotometer was used using Epoch 2 (Bio Tek Instruments, Winooski, VT, USA). Visioline VL-650 was obtained from Courage & Khazaka Electronic GmbH (Cologne, Germany). The GPSKIN barrier light was purchased from GPSkin (Seoul, Republic of Korea). An optical microscope was used (BX61, Tokyo, Japan) and photographed using a DP80 microscope (Olympus). The CFX Connect Real-Time PCR Detection System was purchased from Bio-Rad (Hercules, CA, USA). Chemiluminescence images were acquired using a LuminoGraph III Lite (WSE-6370, Atto, Tokyo, Japan).

All reagents were purchased from Sigma Aldrich (St. Louis, MO, USA) unless otherwise indicated. De Man, Rogosa, and Sharpe (MRS) broth were obtained from Becton Dickinson and Company (Franklin Lakes, NJ, USA). Dulbecco’s modified Eagle’s medium (DMEM), fetal bovine serum (FBS), and penicillin/streptomycin (P/S) were purchased from Gibco (Gaithersburg, MD, USA). Dulbecco’s phosphate-buffered saline (DPBS) purchased from Welgene (Gyeongsan-si, Republic of Korea). To perform real-time polymerase chain reaction, NucleoZol (MACHEREY-NAGEL, Gutenberg, Hoerdt, France) and Maxime RT PreMix (iNtRON, Seongnam-si, Republic of Korea) were used. The contents of pro-collagen type 1 were analyzed using a Procollagen Type I C-peptide (PIP) enzyme-linked immunosorbent assay (EIA) Kit (Takara Bio Inc., Shiga, Japan). AmfiSure qGreen Q-PCR Master Mix, radioimmunoprecipitation (RIPA), Bradford Assay Kit, protease and phosphatase inhibitor cocktail, polyvinylidene fluoride (PVFD) membrane, goat anti-rabbit and anti-mouse IgG(H + L)-HRP, and enhanced chemiluminescence (ECL) were obtained from GenDEPOT (Katy, TX, USA). Primary antibodies were purchased from Cell Signaling Technology (Danvers, MA, USA) and Santa Cruz Biotechnology (Dallas, CA, USA).

### 2.2. Preparation Sample of Bi. lactis MG741

*Bifidobacterium animalis* ssp. *lactis* (*Bi. lactis*) MG741 was isolated from human infants and laboratory-maintained in MEDIOGEN (Jecheon, Republic of Korea). *Bi. lactis* MG741 (NCBI GenBank number: MN069035.1; originated from infant feces) was cultivated in MRS broth with GasPak EZ (Becton, Dickinson, and Company). Cell-free supernatant (CFS) of *Bi. lactis* MG741 for in vitro assays was prepared as previously described [18]. For the in vivo study, the grown *Bi. lactis* MG741 was freeze-dried (20 mTorr, –40 ℃, 1 d) and powdered *Bi. lactis* MG741 was mixed with maltodextrin to obtain a dilution (1 × 10^11^ CFU/g). 

### 2.3. In Vitro Study

Hs68 fibroblasts (ATCC, Manassas, MD, USA) were used in the experiments. The fibroblasts were grown in DMEM containing 10% FBS and 1% P/S in 5% CO_2_ and at 37 ℃ in a humidified atmosphere. The fibroblasts were transferred every two days with fresh growth media.

#### 2.3.1. Cytotoxicity

The cell viability of the CFS from *Bi. lactis* MG741 cells was estimated using the 3-(4,5-dimethylthiazol-2-yl)-2,5-diphenyltetrazolium bromide (MTT) assay. Fibroblasts were seeded in a 96-well plate (1 × 10^4^ cells/well) and incubated overnight. The CFS was treated with 5 and 10% (*v*/*v* in DMEM) in each well for 24 h. After replacing the medium with DPBS, followed by exposure to UVB (30 mJ/cm^2^) using a UV crosslinker, the fibroblasts were incubated with CFS in media for 24 h. The MTT solution (0.25 mg/mL) was added and incubated for 2 h. The medium was discarded and dimethyl sulfoxide (DMSO, 150 μL) was added to dissolve the formazan crystals. Absorbance was measured at 550 nm using a microplate spectrophotometer.

#### 2.3.2. Determination of PIP by EIA

PIP was measured according to the manufacturer’s instructions (Takara Bio, Inc.). For evaluation of PIP, the fibroblasts (8 × 10^4^ cells/well) in a 24-well plate were incubated with or without 10% CFS for 24 h. After exposure to UVB (30 mJ/cm^2^), cells were incubated with CFS for 24 h. PIP content was measured using a microplate spectrophotometer at 450 nm and normalized to protein (μg/μL) using the Bradford assay.

### 2.4. Animals Study

The animal study protocol for the in vivo study was approved by the Ethics Committee of ChemOn Inc., Gyeonggi-do, Republic of Korea (approval number, 2022-0017). Five-week-old female specific pathogen-free (SPF) hairless (HR-1) mice were used in this study (Central Lab. Animal Inc., Seoul, Republic of Korea). All mice were housed in an environment with a controlled temperature of 22 ± 3 °C, a humidity of 55 ± 15%, and illuminance of 150–300 Lux under a 12 h light/dark cycle. During the experimental period, the diet (Teklad certified irradiated global 18% protein rodent diet, 2918C, ENVIGO, Indianapolis, IN, USA) and drinking water were fed by free intake. 

#### 2.4.1. Experimental Design

After 1-week of acclimation, animals were divided by randomized complete block design into three groups (*n* = 6 per group), as follows: (1) normal (no-UVB) group, (2) UVB group (60 to 240 mJ/cm^2^), (3) UVB + *Bi. lactis* MG741 strain (1 × 10^9^ CFU/head/day). UVB irradiation was gradually increased from 60 to 240 mJ/cm^2^ for 12 weeks. The protocol used for the animal model induced by UVB is shown in Figure 1. The mice were administered *Bi. lactis* MG741 using oral gavage. The normal and only UVB-exposed group was administered maltodextrin (0.01 g/head/day). All treatment solutions were freshly dissolved on the experimental days in phosphate-buffered saline (PBS). Body weights were measured at one-week intervals during the 12-week experimental period (Appendix A).

#### 2.4.2. Determination of Wrinkle Formation

For the measurement of wrinkle formation, the skin impression (replica) of the dorsal skin was prepared. At 12 weeks, a replica of each mouse was produced to estimate the total wrinkle area (mm^2^), mean length (mm), mean depth (μm), and maximum wrinkle depth (μm). 

#### 2.4.3. Histopathological and Skin Hydration Examination

A part of the dorsal skin was fixed with 10% neutral buffered formalin and embedded in paraffin blocks. Paraffin blocks were cut into sections and stained with hematoxylin and eosin (H&E). The sections were examined under an optical microscope and photographed. The epidermal thickness, stained with H&E, was measured at 100× magnification. Trans-epidermal water loss (TEWL) and hydration of mouse dorsal skin were determined using GPSKIN barrier light at 12 weeks.

### 2.5. Quantitative RT-PCR (qRT-PCR)

Hs68 fibroblasts were seeded at 3 × 10^5^ cells/well in 6-well plates and treated with or without 10% CFS for 24 h. After replacing the medium with DPBS, followed by exposure to UVB (30 mJ/cm^2^), the fibroblasts were incubated with CFS in media for 24 h. The total RNA of fibroblasts and dorsal skin tissues was extracted using NucleoZol, and mRNA was synthesized to cDNA using Maxime RT PreMix, according to the manufacturer’s instructions [19]. qPCR was performed using the AmfiSure qGreen Q-PCR Master Mix using the CFX Connect system. Primers were designed using Primer-BLAST as shown in Appendix A (NCBI, Bethesda, MD, USA). The threshold cycle (Ct) value for each gene was normalized to glyceraldehyde-3-phosphate dehydrogenase (*GAPDH*). The results were analyzed with CFX Maestro software 2.3 for Windows, provided by Bio-Rad.

### 2.6. Western Blotting

Whole fibroblast lysates and dorsal skin tissue were prepared in a RIPA buffer containing protease and phosphatase inhibitors. Total protein content was determined using the Bradford assay. The proteins were separated by sodium dodecyl sulfate-polyacrylamide gel electrophoresis using 10% acrylamide gel and transferred to PVDF membranes. Membranes were incubated with primary antibodies (MMP-1, MMP-3, p-extracellular signal-regulated kinase (ERK), ERK, p-c-Jun N-terminal kinase (JNK), JNK, p- p38 mitogen-activated protein kinases (p38), p38, p-transcription factor Jun (c-Jun), c-Jun, p-c-Fos, c-Fos, p-NF-κB p65, NF-κB p65, and β-actin) for 18–24 h and secondary antibodies for 1 h. The blots were visualized using an ECL reagent and a chemiluminescence system. Quantitative analysis was performed using the ImageJ software (National Institutes of Health, Bethesda, MD, USA).

### 2.7. Statistics

All experimental results are presented as the mean ± standard error (SE). The statistical significance of differences in this study was calculated using Student’s t-test (Prism 5.02 GraphPad Software, San Diego, CA, USA).

## 3. Results

### 3.1. Protective Effect of Bi. lactis MG741 on UVB-Exposed Hs68 Fibroblasts

To investigate the protective effect in Hs68 fibroblasts, we performed MTT assay (Figure 2a). CFS (10%) from *Bi. lactis* MG741 proliferated Hs68 fibroblasts up to 1.55-fold of no-UVB control, and was markedly protected against UVB-induced cell death (1.56-fold of UVB control). As a result of the energy of UVB exposure (15, 30, 50, and 100 mJ/cm^2^) that causes the death of fibroblasts, they exhibited a cytotoxicity of ~30% at a minimum and morphology change on 30 mJ/cm^2^ (Appendix A). Thus, 30 mJ/cm^2^ UVB irradiation was used in subsequent experiments. *Bi. lactis* MG741 affected PIP degradation in UVB-exposed Hs68 fibroblasts (Figure 2b). CFS (10%) from *Bi. lactis* MG741 increased PIP content (1.17-fold) in fibroblasts and significantly enhanced PIP content up to 2.04-fold in UVB-exposed Hs68 fibroblasts.

### 3.2. Bi. lactis MG741 Exerts Rreduction of mRNA and Protein Expression Related to Wrinkles on UVB-Exposed Hs68 Fibroblasts

Exposure to UVB promoted factors involved in wrinkle formation in fibroblasts by expressing MMPs. However, CFS (10%) from *Bi. lactis* MG741 inhibited the mRNA and protein expressions of MMP-1, and MMP-3, respectively, in UVB-exposed Hs68 fibroblasts (Figure 3a,b). As a result of MAPKs, which activated MMPs, Western blotting revealed that UVB exposure upregulated p-ERK (1.54-fold), p-JNK (1.82-fold), and p-p38 (2.08-fold) in fibroblasts. *Bi. lactis* MG741 reduced the phosphorylation of ERK and JNK in UVB-exposed Hs68 fibroblasts, shown in Figure 3c.

### 3.3. Bi. lactis MG741 Modulates Skin Changes on UVB-Exposed Dorsal Skin of HR-1 Mice 

The effects of *Bi. lactis* MG741 on skin changes in UVB-exposed dorsal skin of HR-1 mice was performed, as shown in Figure 1. As shown in Figure 4, the increase in wrinkle formation, including total wrinkle area (2.85-fold), mean length (1.85-fold), mean depth (1.15-fold), and maximum wrinkle depth (2.05-fold), was shown by UVB, compared with non-exposed HR-1 mice. By contrast, the administration of *Bi. lactis* MG741 ameliorated the wrinkle formation on UVB-exposure in, respectively HR-1 mice. Among them, the total wrinkle area was markedly decreased by *Bi. lactis* MG741 (0.68-fold increase in UVB-exposed dorsal skin of HR-1 mice).

Though H&E staining, UVB exposure in HR-1 mice resulted in marked thickening of the dorsal skin (3.39-fold of non-exposure; Figure 5a,b); however, *Bi. lactis* MG741 reduced the epidermal thickness up to 0.82-fold in UVB-exposed HR-1 mice. As shown in Figure 5c,d, UVB irradiation of the dorsal skin of HR-1 mice induced increase in TEWL (8.92-fold, significantly) and loss of skin hydration (0.91-fold) compared with non-exposure. *Bi. lactis* MG741 mitigated these changes caused by UVB (in TEWL, 0.82-fold; in loss of skin hydration, 1.08-fold). 

### 3.4. Effect of Bi. lactis MG741 on Protein Expression Related to Skin Aging on UVB-Exposed Dorsal Skin of HR-1 Mice

The results of Western blot analysis indicated that the protein expressions of MMP-1 (1.92-fold) and MMP-3 (1.73-fold) were clearly enhanced by UVB exposure in the dorsal skin of HR-1 mice (Figure 6a). The protein expression of MMP-3 was decreased by administration of *Bi. lactis* MG741; however, the MMP-1 expression did not change. As shown in Figure 6b,c, MAPKs (p-ERK, 2.22-fold; pJNK, 1.35-fold; p-p38, 3.13-fold), and AP-1 (p-c-Jun, 1.28-fold; p-c-Fos, 1.56-fold) were also markedly overexpressed in the UVB-exposed dorsal skin of HR-1 mice compared with non-exposed mice. Conversely, *Bi. lactis* MG741 inhibited the protein expression of p-ERK, p-p38, and p-c-Fos, with no change in p-JNK and p-c-Fos.

### 3.5. Bi. lactis MG741 Downregulates Factors Related to Inflammationon on UVB-Exposed Dorsal Skin of HR-1 Mice

The mRNA expressions of *IL-6* and *TNFα* were significantly increased by UVB irradiation up to 4.89- and 6.90-fold of non-exposure, respectively, whereas *Bi. lactis* MG741 supplementation significantly reduced these factors (*IL-6*, 0.17-fold; *TNFα*, 0.23-fold) compared with the UV-exposed control group (Figure 7a). Moreover, protein expression of p-NF-κB analyzed by Western blot increased up to 1.75-fold compared with that in the dorsal skin of HR-1 mice exposed to UVB irradiation. The protein expression of p-NF-κB, increased by UVB exposure, were improved by the administration of *Bi. lactis* strain MG741 (Figure 7b).

## 4. Discussion

Among the photons irradiated from the sun, UV is divided into UVA (320–400 nm), UVB (280–320 nm), and UVC (100–280 nm) [6]. It is known that most UVB is absorbed by the ozone layer, but this amount has increased due to environmental pollution [20]. Long-term UVB irradiation has been reported to increase the decomposition of elastin and collagen in skin aging (such as rough skin, suppression of skin elasticity, and skin wrinkles) and/or has a detrimental effect [21]. Therefore, as the amount of UVB irradiation increases, the effect on the skin is expected to increase. Thus, protective agents have been studied. Recently, probiotics, not synthetic substances, has been studied as alternative products with skin protection effects. It is reported that *L. plantarum* HY7714 relieves skin wrinkles and improves elasticity by controlling skin moisture and MMPs [22,23,24]. In *Bifidobacterium* genera, *Bi. breve* B-3 and *Bi. longum* suppresses skin wrinkles, TEWL, and epidermal thickening by reducing inflammation-related proteins and MMPs [25,26]. In a previous our study, the *Bi. lactis* MG741 exerts antioxidant effects by increasing catalase and reducing lipid peroxidation in cell and animal models [18]. UVB generates ROS in skin, including hydroxyl radicals, superoxide anions, and singlet oxygen [16]. Catalase (antioxidant enzymes), and prevention of lipid peroxide exhibit inhibition of skin aging and protective effects on the skin [27]. Based on these data, we investigated the anti-photoaging mechanism of *Bi. lactis* MG741 in vitro and in vivo. 

Skin aging due to UVB shows histological changes, such as an increase in skin thickness and a decrease in skin moisture [28,29]. Fibroblasts, primarily found in the dermis, protect against external damage to the skin by producing an extracellular matrix through granulation tissue formation [9,30]. The extracellular matrix produced by fibroblasts retains moisture and plays an essential role in balancing the skin hydration [31]. Thus, fibroblasts that function normally can maintain the strength and indirectly moisture of the skin. In this study, it was confirmed that *Bi. lactis* MG741 attenuated skin thickening by inhibiting UVB-induced apoptosis in proliferating fibroblasts and affected TEWL and hydration by the proliferation of fibroblasts. 

Chronic UVB exposure accelerates the degradation of collagen and causes cell death, reducing skin strength and elasticity [21]. Most skin collagen is collagen type 1, a protein produced by the synthesis and enzymatic reaction of PIP, and is mainly distributed in fibroblasts [32]. MG741 inhibited PIP degradation in UVB-exposed fibroblasts by increasing PIP secretion. When collagen, a component of the dermis, is damaged by UVB, its structure is deformed exacerbating the formation of wrinkles that are directly or indirectly affected by a decrease in skin elasticity [33]. In our study, *Bi. lactis* MG741 reduced wrinkle formation in UVB-exposed HR-1 mice. MMPs have 20 isoforms and are known to mainly function in the decomposition of the extracellular matrix and remodeling of tissues [34]. Among them, MMP-1 and MMP-3 are major factors that break down collagen type 1 and are initially enhanced by UVB irradiation in the fibroblasts [35]. The factor most affected by MAPKs (ERK, JNK, and p38) is AP-1, which includes c-Jun and c-Fos [36]. Phosphorylation of JNK/P38 stimulates c-Jun, whereas phosphorylation of ERK activates c-Fos [37]. Activated AP-1 expresses MMP-1 and MMP-3 and eventually degrades the collagen [36]. In cellular and animal models, *Bi. lactis* MG741 was identified as having a molecular mechanism to reduce MMP-3 through the inhibition of p-c-Fos, commonly activated by p-ERK. NF-κB is an important transcription factor that regulates immunity and various functions related to aging, energy homeostasis, and cell regeneration [38]. UVB exposure induces the production of *IL-6* and *TNFα*, triggering a cascade of NF-κB reactivation [28,39]. Moreover, UVB-activated MAPK stimulates NF-κB phosphorylation to increase MMPs expression, accelerating skin aging by inducing an extracellular matrix degradation [39]. *Bi. lactis* MG741 exerted efficacy in inflammation and aging-related wrinkle formation by downregulating pro-inflammatory cytokines (*IL-6* and *TNF-α*) and the protein expression of p-NF-κB.

This study confirmed that *Bi. lactis* MG741 mediates the MMP/AP-1/NFκB signaling pathway to protect the skin from UVB and prevent skin aging in cell and animal models. However, the in vitro and in vivo results suggest that *Bi. lactis* MG741 strain is a potential candidate for clinical trials. Further studies are required to fully elucidate the use of *Bi. lactis* MG741 in humans.

## 5. Conclusions

In summary, this study demonstrated that *Bi. lactis* MG741 improved pro-collagen type 1 and skin wrinkle formation by modulating MMP/AP-1 in UVB-exposed fibroblasts and HR-1 mice. In addition, UVB-induced skin inflammation alleviated by *Bi. lactis* MG741 via inhibition of the NF-κB signaling pathway. The results of this study suggest that *Bi. lactis* MG741 can provide a functional food as an anti-photoaging agent.

## Figures and Tables

**Figure 1 microorganisms-10-02343-f001:**
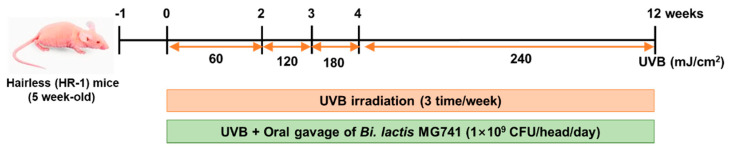
Animal experiment procedure of UVB irradiation on HR-1 mice. After adaptation, the dorsal skin of the HR-1 mice was irradiated to UVB (290 to 320 nm) at one minimal erythema dose (MED, 60 mJ/cm^2^) for two weeks, two MED (120 mJ/cm^2^) for one week, three MED (180 mJ/cm^2^) for one week, and four MED (240 mJ/cm^2^) for the last eight weeks.

**Figure 2 microorganisms-10-02343-f002:**
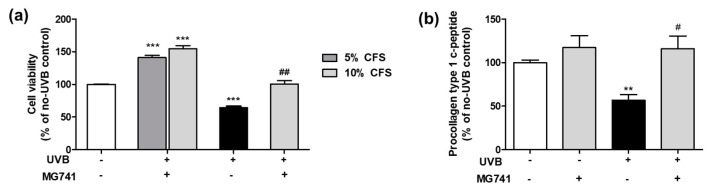
Effects of CFS from *Bi. lactis* MG741 on (**a**) cell viability and (**b**) procollagen type 1 c-peptide (PIP) contents in UVB-exposed Hs68 fibroblasts. Data are indicated as means ± standard error (SE, *n* = 3). ** *p* < 0.01, and *** *p* < 0.001 vs. no-UVB control; ^#^
*p* < 0.05, and ^##^
*p* < 0.01 vs. UVB-exposed control.

**Figure 3 microorganisms-10-02343-f003:**
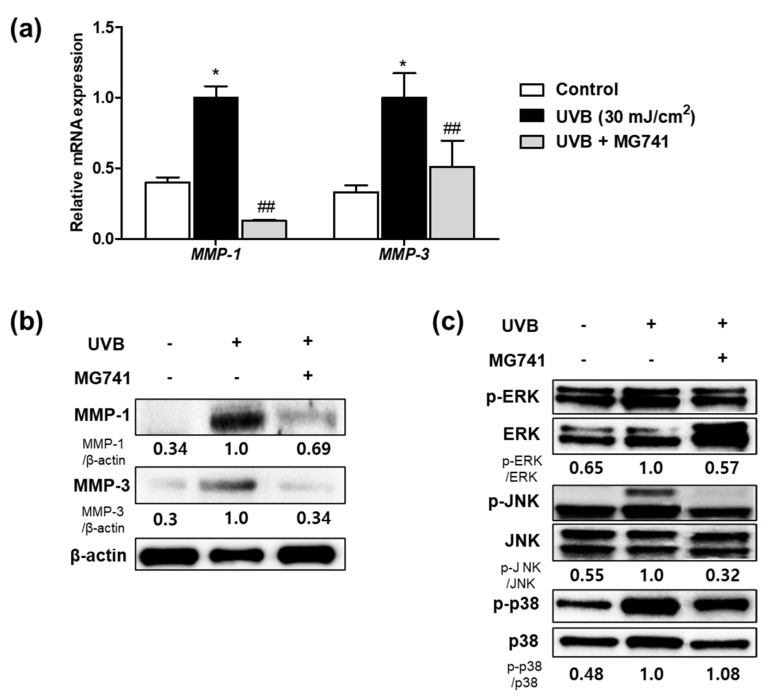
The mRNA and protein expression of matrix metalloproteinases (MMPs) and mitogen-activated protein kinases (MAPKs) in CFS from *Bi. lactis* MG741 treatment with UVB-exposed Hs68 fibroblasts. (**a**) The mRNA expression levels of MMPs in UVB (30 mJ/cm^2^)-exposed Hs68 fibroblasts treated with CFS (10%) from *Bi. lactis* MG741. (**b**,**c**) The Western blot images were measured following treatment with CFS and UVB. Fibroblasts were treated with CFS (10%) *Bi. lactis* MG741 for 24 h. Subsequently, the cells were exposed to UVB (30 mJ/cm^2^) and it was further cultured in media containing CFS for 48 h for MMPs, and 24 h for MAPKs. The ratio of each protein was shown below the blotting images and indicated to fold of UVB-exposed control. Data are indicated as means ± SE (*n* = 3). * *p* < 0.05 vs. no-UVB control; ^##^
*p* < 0.01 vs. UVB-exposed control.

**Figure 4 microorganisms-10-02343-f004:**
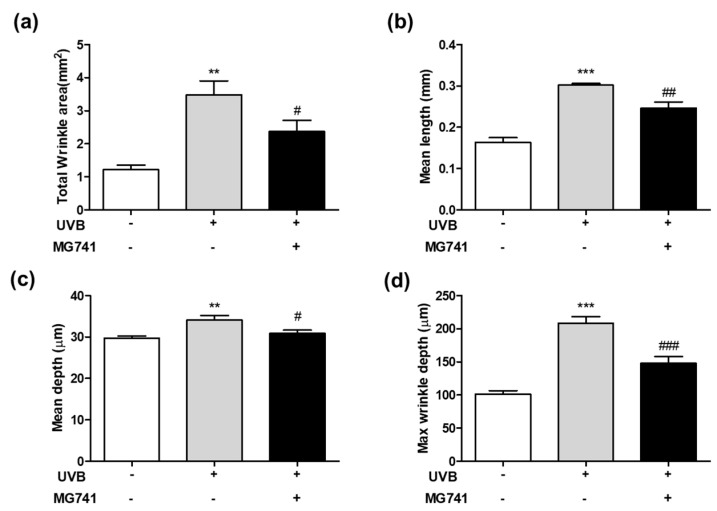
Effect of *Bi. lactis* MG741 on skin wrinkle formation such as (**a**) total wrinkle area, (**b**) mean length, (**c**) mean depth and (**d**) max wrinkle depth in UVB-exposed HR-1 mice. HR-1 mice were treated with *Bi. lactis* MG741 (1 × 10^9^ CFU/head/day) orally and UVB irradiation on dorsal skin until 12 weeks. Data are indicated as means ± SE (*n* = 6). ** *p* < 0.01, and *** *p* < 0.001 vs. no-UVB control; ^#^
*p* < 0.05, ^##^
*p* < 0.01, and ^###^
*p* < 0.001 vs. UVB-exposed control.

**Figure 5 microorganisms-10-02343-f005:**
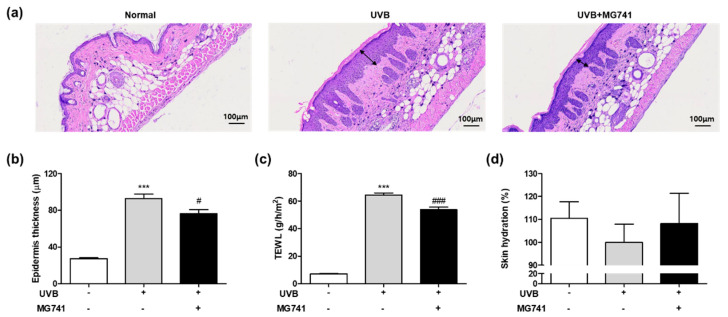
Histological examination and skin hydration of UVB-exposed dorsal skin tissue of HR-1 mice treated with *Bi. lactis* MG741. The animal experiment procedure is the same as described in the legend in Figure 4. (**a**) The images of dorsal skin sections were stained by hematoxylin and eosin (H&E) using a microscope at 100× magnification. Double arrows indicate the epidermis. At 12 weeks, the dorsal skins were collected to measure (**b**) the epidermis thickness (based on H&E stain), (**c**) TEWL, and (**d**) skin hydration. Data are indicated as means ± SE (*n* = 6). *** *p* < 0.001 vs. no-UVB control; ^#^
*p* < 0.05, and ^###^
*p* < 0.001 vs. UVB-exposed control.

**Figure 6 microorganisms-10-02343-f006:**
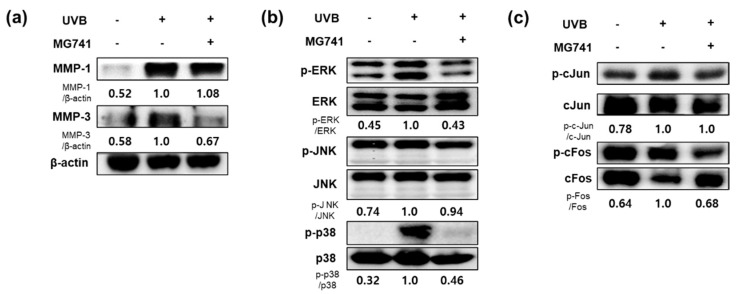
The protein expressions of wrinkle-related factors in *Bi. lactis* MG741 administration with UVB-exposed dorsal skin tissue of HR-1 mice. The animal experiment procedure is the same as described in the legend in Figure 4. (**a**) MMPs, (**b**) MAPKs, and (**c**) activator protein 1 (AP-1) expressions of mouse dorsal skin were analyzed by Western blotting. The ratio of each protein was shown below the blotting images and indicated to fold of UVB-exposed control. Data are indicated as means ± SE (*n* = 6).

**Figure 7 microorganisms-10-02343-f007:**
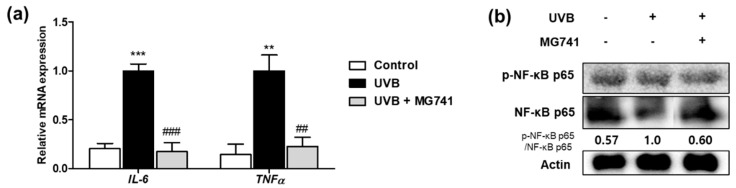
Effect of *Bi. lactis* MG741 on inflammation-related factors in UVB-exposed HR-1 mice. (**a**) The mRNA expressions of *IL-6* and *TNFα* were measured by qRT-PCR. (**b**) The ratio of p-NF-κB/NF-κB was shown below the blotting images and indicated to fold of UVB-exposed control. Data are indicated as means ± SE (*n* = 6). ** *p* < 0.01, and *** *p* < 0.001 vs. no-UVB control; ^##^
*p* < 0.01, and ^###^
*p* < 0.001 vs. UVB-exposed control.

## Data Availability

Not applicable.

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
