# Peer review of "Protective Effect of *Bifidobacterium animalis* subs. *lactis* MG741 as Probiotics against UVB-Exposed Fibroblasts and Hairless Mice"

_microorganisms, 2022, doi:10.3390/microorganisms10122343_

Round 1
Reviewer 1 Report
The authors analyzed the influence of probiotic bacteria on skin aging. The study design is interesting and novel. However, the authors should provide a picture of irradiated mouse skin to demonstrate wrinkle formation of exposed skin together with non-irradiated skin.
Author Response
Authors’ response (in blue) to Reviewer#1’s comments (in italic black):
Comments and Suggestions for Authors
The authors analyzed the influence of probiotic bacteria on skin aging. The study design is interesting and novel. However, the authors should provide a picture of irradiated mouse skin to demonstrate the wrinkle formation of exposed skin together with non-irradiated skin.
Response:
We thank the anonymous reviewer for thoroughly reading our manuscript and we agree with your comments. We tried to insert a replica picture, but the picture was not clear, so we deleted it. However, we will add it to the supplementary if you request. Please check the attachment.

Reviewer 2 Report
In this research article, the authors investigated the ability of metabolites and viable Bifidobacterium animalis ssp. lactis MG741 to reverse UVB-induced changes in gene expression in vitro and in vivo, respectively. Additionally, they studied the ability of administration of viable cells to reduce wrinkle formation and symptoms of UVB-induced aging in hairless mice. The authors conclude that treatments with this strain reversed the detrimental effects of UVB exposure, including wrinkle formation and loss of skin hydration, by regulating the phosphorylation of key signaling molecules, including NF-kB subsequently leading to the downregulation of MMPs and of pro-inflammatory factors. Overall, this work is comprehensive and coherent, however, some minor points require attention:
· The use of language should be edited as passages in the manuscript are unclear (e.g. Lines 9-10, 182-184, 196 etc).
· The definition of probiotics is not the one included in lines 41-42. Please revise using the one given by Hill et al., 2014.
· More details about the treatments should be included in the gene expression analyses section in materials and methods (Lines 156-175) – the treatment timepoints should be added in the text.
· Line 186: how were the morphology changes recorded and scored?
· Line 198: it is unclear how cell lines can wrinkle after UVB exposure. Please correct this sentence.
· Lines 211-212: were cells continuously exposed to UVB for 24 or 48 hours? Is this physiologically relevant? Additionally, exposure for as little as half a minute can significantly limit cell viability, let alone for 24 or 48 hours.
· Line 235, Figure 5d: no significant reduction in skin hydration percentage is shown in the figure. Please revise the text.
· Lines 271-272: what do the authors mean by “These increases were improved by the administration of Bi. lactis strain MG741 (Figure 7b)”? Please revise this passage.
· Line 316: the word isoforms may be more appropriate rather than “structures”.
Author Response
Authors’ response (in blue) to the Reviewer#2’s comments (in italic black):
Comments and Suggestions for Authors
In this research article, the authors investigated the ability of metabolites and viable Bifidobacterium animalis ssp. lactis MG741 to reverse UVB-induced changes in gene expression in vitro and in vivo, respectively. Additionally, they studied the ability of administration of viable cells to reduce wrinkle formation and symptoms of UVB-induced aging in hairless mice. The authors conclude that treatments with this strain reversed the detrimental effects of UVB exposure, including wrinkle formation and loss of skin hydration, by regulating the phosphorylation of key signaling molecules, including NF-kB subsequently leading to the downregulation of MMPs and of pro-inflammatory factors. Overall, this work is comprehensive and coherent, however, some minor points require attention:
Response:
thank the anonymous reviewer for thoroughly reading our manuscript and providing helpful comments and suggestions. The detailed responses to major points are listed below:
The use of language should be edited as passages in the manuscript are unclear (e.g. Lines 9-10, 182-184, 196 etc).
Response:
Thank you for your comment, we have edited the passages you suggested (Line 9-10, 188-190, 202).
The definition of probiotics is not the one included in lines 41-42. Please revise using the one given by Hill et al., 2014.
Response:
Thank you for your comment, we have revised about the definition of probiotics.
P1, L42 – “According to the World Health Organization (WHO) and Food and Agriculture Organization of the United Nations (FAO), probiotics, including Lactobacillus and Bifidobacterium, are living microorganisms that have health benefit on host, when ad-ministered in adequate amounts.”
More details about the treatments should be included in the gene expression analyses section in materials and methods (Lines 156-175) – the treatment timepoints should be added in the text.
Response:
Thank you for your comment, we have revised about the definition of probiotics.
P4, L160 – “Hs68 fibroblasts were seeded at 3 × 105 cells/well in 6-well plates and treated with or without 10% CFS for 24 h. After replacing the medium with DPBS, followed by ex-posure to UVB (30 mJ/cm2), the fibroblasts were incubated with CFS in media for 24 h.~”
Line 186: how were the morphology changes recorded and scored?
Response:
Thank you for your comment. Referring to Supplementary data Figure S2, cell morphology was confirmed through photographs, and viability was measured through MTT assay to confirm.
Line 198: it is unclear how cell lines can wrinkle after UVB exposure. Please correct this sentence.
Response:
Thank you for your comment, we have revised that passage.
P5, L205 – “Exposure to UVB promoted factors involved in wrinkle formation in fibroblasts by expressing MMPs.”
Lines 211-212: were cells continuously exposed to UVB for 24 or 48 hours? Is this physiologically relevant? Additionally, exposure for as little as half a minute can significantly limit cell viability, let alone for 24 or 48 hours.
Response:
Thank you for your comment, we have revised that passage.
P6, L218 – “Subsequently, the cells were exposed to UVB (30 mJ/cm2) and it was further cultured in media containing CFS for 48 h for MMPs, and 24 h for MAPKs”
Line 235, Figure 5d: no significant reduction in skin hydration percentage is shown in the figure. Please revise the text.
Response:
Thank you for your comment, we agree with your suggestion. We have modified the manuscript accordingly in line 242.
Lines 271-272: what do the authors mean by “These increases were improved by the administration of Bi. lactis strain MG741 (Figure 7b)”? Please revise this passage.
Response:
Thank you for your comment, we have revised that passage.
P8, L278 – “The protein expression of p-NF-κB, increased by UVB exposure, were improved by the administration of Bi. lactis strain MG741 (Figure 7b).”
Line 316: the word isoforms may be more appropriate rather than “structures”.
Response:
Thank you for your comment, we have corrected the manuscript accordingly in page 9, line 322.

Reviewer 3 Report
Review of Microorganism paper entitled Protective Effect of Bifidobacterium animalis ssp. lactis MG741 2 against UVB-exposed Fibroblasts and Hairless Mice
General Review:
The paper entitled “Protective Effect of Bifidobacterium animalis ssp. lactis MG741 2 against UVB-exposed Fibroblasts and Hairless Mice” is a well-written paper that examines important elements of the role of probiotics and also post-biotic Cell-Free Supernatant treatments on mouse skin and human skin cells (Fibroblasts). The title should include the fact that the mouse treatments are not topical but oral, perhaps something to the effect: Protective Effects of Bifidobacterium animalis ssp. lactis MG741 2 Cell-Free Supernatant postbiotic in vitro and oral probiotics against UVB-exposed Fibroblasts and Hairless Mice.
The authors do a good job of differentiating the in vitro work from the animal work and linking the two together (although one is done on human cells and the other is an animal model). The data is clearly presented and easy to follow. The paper is well-referenced and presumably follows the guidelines that the journal requires for such papers. The paper is suitable for publication with a few minor edits as noted below.
The following are some minor improvements that the paper will benefit from:
Introduction
Lines 32-33: “Therefore, it is crucial to develop agents that are effective for photoaging [that] are proven safe [7].
The authors have not proven safety of their treatments necessarily. The paper is principally an examination of efficacy. The sentence above is part of the introduction and so is okay but the authors should not make any assessments of safety that are not borne out by the results of the animal studies.
Line 47: “It has been previously reported that Lactobacillus plantarum, Bi. 46 longum, and Bi. breve [have] shown efficacy…”
Results
Line 196: “3.2. Bi. lactis MG741 [Exerts Reduction] of Factors Related [to] Wrinkle(s) on UVB-Exposed Hs68…”
Line 198: The following statement is incorrect: “Exposure to UVB promoted wrinkles in fibroblasts by expressing MMPs.” It is not possible to promote wrinkles in fibroblasts. The authors need to adjust this sentence to reflect what they actually are measuring using fibroblasts.
Lines 220-221: “…the administration of Bi. lactis MG741 ameliorated UVB-exposure in HR-1 mice.”
Figure 5: The data for the results of skin hydration (Figure 5d) does not appear to show any statistical significance. Yet, within the test (Line 235), the authors suggest that the treatment provides “…loss of skin hydration (0.91-fold) compared with non-exposure.” The authors must make it clear that the data on skin hydration, all measurements, do not show statistical significance. Thus, the statement above should reflect this uncertainty.
Line 264: “Bi. lactis MG741 Downregulates Factors Related [to] Inflammation [on] UVB-Exposed Dorsal…
Discussion
Line 288: “…owing to cost concerns.” There are likely other benefits besides cost which would be reasons for examining the benefits of probiotics. This should be dropped or modified.
Line 294: “UVB generated in skin ROS…” should read “UVB generates ROS in skin…”
Lines 299-301: “Skin aging due to UVB shows histological changes, such as an increase in skin thickness due to keratin fibers and a decrease in skin moisture that hardens with the aging [27,28]”. This sentence is confusing. UVB will increase skin thickness, but this may not be solely due to increases in keratin fibers. Also, “a decrease in skin moisture that hardens with aging” makes no sense. I think the authors mean that a decrease in skin moisture that increases with aging. Please adjust this sentence.
Lines 304-305: “Thus, fibroblasts that function normally can maintain the thickness and moisture of the skin”. Fibroblasts play a somewhat minor role in maintaining skin moisture. The collagen produced by the skin is principally responsible for the strength (not elasticity, Line 310) of the skin. Elastin fibers are principally responsible for skin’s elasticity. The authors need to adjust their discussion around the role of collagen in the skin.
Lines 313-314: “When collagen, a component of the dermis, is damaged by UVB, its structure is deformed affecting skin elasticity…” Again, collagen does not necessarily affect skin elasticity, it effects the skin’s strength.
Author Response
Authors’ response (in blue) to the Reviewer#3’s comments (in italic black):
General Review:
The paper entitled “Protective Effect of Bifidobacterium animalis ssp. lactis MG741 2 against UVB-exposed Fibroblasts and Hairless Mice” is a well-written paper that examines important elements of the role of probiotics and also post-biotic Cell-Free Supernatant treatments on mouse skin and human skin cells (Fibroblasts). The title should include the fact that the mouse treatments are not topical but oral, perhaps something to the effect: Protective Effects of Bifidobacterium animalis ssp. lactis MG741 2 Cell-Free Supernatant postbiotic in vitro and oral probiotics against UVB-exposed Fibroblasts and Hairless Mice. The authors do a good job of differentiating the in vitro work from the animal work and linking the two together (although one is done on human cells and the other is an animal model). The data is clearly presented and easy to follow. The paper is well-referenced and presumably follows the guidelines that the journal requires for such papers. The paper is suitable for publication with a few minor edits as noted below.
The following are some minor improvements that the paper will benefit from:
Response:
thank the anonymous reviewer for thoroughly reading our manuscript and providing helpful comments and suggestions. The detailed responses to minor points are listed below:
Introduction
Lines 32-33: “Therefore, it is crucial to develop agents that are effective for photoaging [that] are proven safe [7].
The authors have not proven the safety of their treatments necessarily. The paper is principally an examination of efficacy. The sentence above is part of the introduction and so is okay but the authors should not make any assessments of safety that are not borne out by the results of the animal studies.
Response:
Thank you for your comment. In our previous study, we evaluated the safety of MG741 through hemolytic and antibiotic resistance tests. We have revised that passage.
P2, L58 – “We have previously demonstrated that Bi. lactis MG741 has antioxidant potential in a tert-butyl hydroperoxide (t-BHP)-induced animal model, and was evaluated for safety as probiotics [18].”
Line 47: “It has been previously reported that Lactobacillus plantarum, Bi. 46 longum, and Bi. breve [have] shown efficacy…”
Response:
Thank you for your comment, we have corrected the manuscript accordingly in page 2, line 49.
Results
Line 196: “3.2. Bi. lactis MG741 [Exerts Reduction] of Factors Related [to] Wrinkle(s) on UVB-Exposed Hs68…”
Response:
Thank you for your comment, we have corrected the manuscript accordingly in page 5, line 203.
Line 198: The following statement is incorrect: “Exposure to UVB promoted wrinkles in fibroblasts by expressing MMPs.” It is not possible to promote wrinkles in fibroblasts. The authors need to adjust this sentence to reflect what they actually are measuring using fibroblasts.
Response:
Thank you for your comment, we have revised that passage.
P5, L205 – “Exposure to UVB promoted factors involved in wrinkle formation in fibroblasts by expressing MMPs.”
Lines 220-221: “…the administration of Bi. lactis MG741 ameliorated UVB-exposure in HR-1 mice.”
Response:
Thank you for your comment, we have revised that passage.
P6, L227 – “By contrast, the administration of Bi. lactis MG741 ameliorated the wrinkle formation on UVB-exposure in respectively HR-1 mice”
Figure 5: The data for the results of skin hydration (Figure 5d) does not appear to show any statistical significance. Yet, within the test (Line 235), the authors suggest that the treatment provides “…loss of skin hydration (0.91-fold) compared with non-exposure.” The authors must make it clear that the data on skin hydration, all measurements, do not show statistical significance. Thus, the statement above should reflect this uncertainty.
Response:
Thank you for your comment, we agree with your suggestion. We have modified the manuscript accordingly in line 242.
Line 264: “Bi. lactis MG741 Downregulates Factors Related [to] Inflammation [on] UVB-Exposed Dorsal…
Response:
Thank you for your comment, we have corrected the manuscript accordingly in page 6, line 271.
Discussion
Line 288: “…owing to cost concerns.” There are likely other benefits besides cost which would be reasons for examining the benefits of probiotics. This should be dropped or modified.
Response:
Thank you for your comment, we have deleted the sentence.
Line 294: “UVB generated in skin ROS…” should read “UVB generates ROS in skin…”
Response:
Thank you for your comment, we have corrected the manuscript accordingly in page 9, line 301.
Lines 299-301: “Skin aging due to UVB shows histological changes, such as an increase in skin thickness due to keratin fibers and a decrease in skin moisture that hardens with the aging [27,28]”. This sentence is confusing. UVB will increase skin thickness, but this may not be solely due to increases in keratin fibers. Also, “a decrease in skin moisture that hardens with aging” makes no sense. I think the authors mean that a decrease in skin moisture that increases with aging. Please adjust this sentence.
Response:
Thank you for your comment, we have corrected the manuscript accordingly in page 9, line 306.
Lines 304-305: “Thus, fibroblasts that function normally can maintain the thickness and moisture of the skin”. Fibroblasts play a somewhat minor role in maintaining skin moisture. The collagen produced by the skin is principally responsible for the strength (not elasticity, Line 310) of the skin. Elastin fibers are principally responsible for skin’s elasticity. The authors need to adjust their discussion around the role of collagen in the skin.
Response:
Thank you for your comment. In ref. [31], HA in the extracellular matrix has water-holding properties. Since extracellular matrix are produced by fibroblasts, fibroblasts are indirectly involved in hydration. Thus, we have revised that passage.
P9, L311 – “Thus, fibroblasts that function normally can maintain the strength and indirectly moisture of the skin.”
Lines 313-314: “When collagen, a component of the dermis, is damaged by UVB, its structure is deformed affecting skin elasticity…” Again, collagen does not necessarily affect skin elasticity, it effects the skin’s strength.
Response:
Thank you for your comment, we have revised that passage.
P9, L319 – “When collagen, a component of the dermis, is damaged by UVB, its structure is deformed exacerbating the formation of wrinkles that are directly or indirectly affected by a decrease in skin elasticity [33].”
